# Potential Antifungal Effect of Copper Oxide Nanoparticles Combined with Fungicides against *Botrytis cinerea* and *Fusarium oxysporum*

**DOI:** 10.3390/antibiotics13030215

**Published:** 2024-02-26

**Authors:** Javiera Parada, Gonzalo Tortella, Amedea B. Seabra, Paola Fincheira, Olga Rubilar

**Affiliations:** 1Biotechnological Research Center Applied to the Environment (CIBAMA-BIOREN), Faculty of Engendering and Science, Universidad de La Frontera, Temuco 4811230, Chile; gonzalo.tortella@ufrontera.cl (G.T.); paola.fincheira@ufrontera.cl (P.F.); olga.rubilar@ufrontera.cl (O.R.); 2Chemical Engineering Department, Faculty of Engendering and Science, Universidad de La Frontera, Temuco 4811230, Chile; 3Center for Natural and Human Sciences, Universidade Federal do ABC, Santo André 09210-580, Brazil; amedea.seabra@ufabc.edu.br

**Keywords:** copper oxide nanoparticles, antifungal, synergism, fungicide

## Abstract

Copper oxide nanoparticles (NCuO) have emerged as an alternative to pesticides due to their antifungal effect against various phytopathogens. Combining them with fungicides represents an advantageous strategy for reducing the necessary amount of both agents to inhibit fungal growth, simultaneously reducing their environmental release. This study aimed to evaluate the antifungal activity of NCuO combined with three fungicide models separately: Iprodione (IPR), Tebuconazole (TEB), and Pyrimethanil (PYR) against two phytopathogenic fungi: *Botrytis cinerea* and *Fusarium oxysporum.* The fractional inhibitory concentration (FIC) was calculated as a synergism indicator (FIC ≤ 0.5). The NCuO interacted synergistically with TEB against both fungi and with IPR only against *B. cinerea*. The interaction with PYR was additive against both fungi (FIC > 0.5). The *B. cinerea* biomass was inhibited by 80.9% and 93% using 20 mg L^−1^ NCuO + 1.56 mg L^−1^ TEB, and 40 mg L^−1^ NCuO + 12 µg L^−1^ IPR, respectively, without significant differences compared to the inhibition provoked by 160 mg L^−1^ NCuO. Additionally, the protein leakage and nucleic acid release were also evaluated as mechanisms associated with the synergistic effect. The results obtained in this study revealed that combining nanoparticles with fungicides can be an adequate strategy to significantly reduce the release of metals and agrochemicals into the environment after being used as antifungals.

## 1. Introduction

The negative consequences of global warming on agriculture have been evident in recent years. Continuous and severe temperature fluctuations have resulted in decreased arable lands and more frequent recurrence of plant diseases—for instance, those caused by fungal phytopathogens, such as *Botrytis cinerea* and *Fusarium oxysporum* [1]. To counteract these effects, farmers have overused agrochemicals, whose production has contributed considerably to the emission of greenhouse gases and has accelerated the extinction rate of some plants and animals [2,3]. The environmental cost of the uncontrolled use of pesticides also implies a negative impact on human health, mainly due to diseases associated with farmers’ lack of awareness of manual hand spraying of pesticides. Indeed, the excessive use of pesticides has resulted in the induction of resistance genes on target pests, which have transferred from agriculture to human-health-concern strains [4,5,6]. Many governments have modified their legislation to reduce or ban the use of some pesticides [7,8]. However, it is still assumed that pesticides play a pivotal role in crop production as the worldwide population grows: up to 10 billion by 2050, as expected by a report by the Food and Agriculture Organization of the United Nations (FAO, 2017) [9].

The discovery of innovative alternatives that are less harmful to the environment has been one of the main challenges of the scientific community. Thus, formulations based on metal nanoparticles have emerged as a promising strategy due to their small size (~1–100 nm), facilitating better solubility and reactivity of metal ions. Among metals, copper is a recognized broad-spectrum antimicrobial agent usually present in agrochemical products. Currently, the formulation of copper and copper oxide nanoparticles (NCuO) is the focus of many studies due to their reported better antimicrobial effect than copper salts [10,11,12]. Also, NCuO share similar antimicrobial activity to other noble metals such as silver and gold; however, they are considered cost-effective due to their cheaper and easier production and their more abundant presence in nature [13,14]. Despite the demonstrated efficacy of NCuO as antimicrobials, their use is still controversial due to the potential ecological risks associated with metal nanoparticles. In fact, the use of copper-based fungicides has been limited due to their reported negative effects on soil microbial composition [15,16,17,18].

Combining metal nanoparticles (MNPs) with other antimicrobial agents (e.g., antibiotics, essential oils, or other metals) has been proposed as one of the most interesting strategies to avoid or reduce their harmful effects once released into the environment [19,20]. In this sense, some studies have evidenced that the combination of MNPs, such as silver or zinc oxide, with fungicides has provoked additive and synergistic interactions, implying that the antifungal effect has been higher than that of either agent alone [21,22]. In particular, the interaction of NCuO with fungicides remains poorly explored in this context.

The main objective of the present work was to determine whether the antifungal activity of NCuO can improve in the presence of other fungicides. This combination would facilitate maintaining the antifungal activity of both agents by using a lower amount and also reduce the environmental input of copper and fungicides. *Botrytis cinerea* and *Fusarium oxysporum* were used as phytopathogenic test models, known to cause severe diseases on many crops and economic losses worldwide [23,24]. Also, three fungicides commonly used to control these fungal strains and others were selected: Iprodione, Tebuconazole, and Pyrimethanil [25,26]. The type of interaction between NCuO and the fungicides (i.e., synergistic, additive, or antagonist) was also evaluated. The findings obtained from this study evidence the feasibility of using combined strategies of copper oxide nanoparticles with fungicides to control phytopathogenic fungi and simultaneously reduce the ecological risks due to the environmental release of metals and agrochemicals.

## 2. Results and Discussion

### 2.1. Checkerboard Assay

According to the results observed on the checkerboard assay (Figure 1), the MIC of the three evaluated fungicides and NCuO decreased when they were used in combinations. As observed, *B. cinerea* was more sensitive than *F. oxysporum* since lower fungicide amounts were necessary to inhibit its growth. However, this difference in fungicide susceptibility can be associated with each fungus’s intrinsic morphological and structural characteristics. For instance, the cell wall composition (i.e., glycoproteins or chitin amount) is known to vary significantly depending on the species, being used to differentiate fungal genera. At the same time, glycoproteins are recognized as the first defense line in fungi. Therefore, a different response to fungicides could be expected among various species [27,28].

As shown in Figure 1a,d, NCuO exerted a synergistic interaction with TEB, reducing 4-fold its MIC against *B. cinerea* (from 1.2 to 0.3 mg L^−1^) and *F. oxysporum* (from 6.25 to 1.56 mg L^−1^). Simultaneously, the MIC of NCuO was 8-fold reduced: from 160 mg L^−1^ alone to 20 mg L^−1^ in combination). Other fungicides belonging to the same group of TEB have exerted a synergistic effect with copper against some fungal species, such as *Elsinoe ampelina, Glomerella cingulata*, and *Physalospora berengeriana* [29,30]. According to the authors, the synergism can be associated with the formation of a complex between copper ions (as Cu^2+^) and the triazolic ligands of the fungicide. However, scarce information exists about how different the antifungal activity can be if copper is present as nanoparticles. In the case of IPR, the interaction with NCuO was synergistic for *B. cinerea* (Figure 1b) and additive for *F. oxysporum* (Figure 1e). Thus, the MIC of IPR against *B. cinerea* was 4-fold reduced (from 48 µg L^−1^ to 12 µg L^−1^), similarly to NCuO (from 160 mg L^−1^ to 40 mg L^−1^). To our knowledge, this is the first evidence of copper oxide as nanoparticle and TEB or IPR, acting synergistically against *B. cinerea*. The effectiveness of other copper nanoparticles (CuNPs) used individually against *F. oxysporum* has been documented. However, higher inhibitory concentrations were previously reported [31,32] compared to that found in this study (20 mg L^−1^ of NCuO in the presence of fungicides). For example, Ashraf et al. [33] reported 91% of *F. oxysporum* mycelial growth inhibition using 350 mg L^−1^ of copper oxide nanoparticles. Lopez-Lima et al. [34] reported an inhibition of 67% using 500 mg L^−1^. Herein, we found a strategy to reduce significantly the amount of copper released to the environment, through the combination of NCuO with fungicides at more reduced concentrations.

### 2.2. Effect of Synergistic Combinations against Mycelial Biomass

In a longer-term experiment than the previous one (10 days), the synergistic combinations obtained from the checkerboard assay were used to quantify the inhibition percentage on mycelial biomass of *B. cinerea* and *F. oxysporum*. This assay was performed because the fungal cell wall composition varies depending on ageing. For instance, at the stage of spore germination *B. cinerea* has demonstrated to be more sensitive than the mycelial stage [35]. Consequently, the antifungal effect of the combinations needs to be tested in this growth stage. As observed in Figure 2, it was confirmed that the combinations also exerted a synergistic inhibitory effect on the growth of mycelium, evidenced by the obtained synergy factor >1.5 for the three cases, according to the Abbot method (Table 1). Significant differences were found between the combinations and each agent alone for both fungi. For instance, the inhibitions of NCuO (40 mg L^−1^) and IPR (12 µg L^−1^) against *B. cinerea* individually were 55.8% and 5.8%, respectively (Figure 2b). However, the inhibition increased up to 93% when these compounds at the same doses were combined. This inhibition was higher than the expected combined inhibition (% CI_exp_) of 58.4%, according to the Abbot equation (Table 1). Indeed, the inhibition became statistically comparable to the effect of the NCuO MIC used alone (160 mg L^−1^), thus significantly reducing the necessary amount of NCuO to inhibit the fungal biomass. Similar tendencies were observed for the rest of the combinations. In fact, a reduced amount of NCuO (20 mg L^−1^) also provoked a synergic effect with TEB (0.3 mg L^−1^) against the same fungi, evidenced by 84.3% inhibition (Figure 3d). This value was slightly lower than the 88.9% inhibition exerted by the MIC of TEB (1.2 mg L^−1^). The synergism was also corroborated on *F. oxysporum* biomass, inhibited by 80.9% using the synergistic combination (20 mg L^−1^ NCuO + 1.56 mg L^−1^ TEB), which was statistically similar to the inhibition of NCuO at 160 mg L^−1^ (86.9%) or TEB at 6.25 mg L^−1^ (83.8%). Other studies have also addressed the synergism between copper oxide nanoparticles and other antimicrobial agents against different microorganisms. For instance, in combination with anthraquinone-2-carboxylic acid (AQ), the MIC of the nanoparticles against different *Staphyloccus* strains was reduced four- and eight-fold. Thus, the authors proposed that combination as a novel strategy to eradicate human pathogens [36]. Potent synergism with the antibiotic amoxiclav against human pathogens (*Proteus mirabilis* and *Staphylococcus aureus*) has been also evidenced. However, the synergism was achieved with different amounts of both agents for the two bacterial strains, which denotes the importance of considering the concentrations and different target organisms in this type of studies) [37].

On the other hand, synergism between other CuNPs at higher doses (300–500 mg L^−1^) and fungicides against *B. cinerea* at in vitro conditions has also been reported [38]. This can be comparable to the dose of Cu(OH)_2_ (commercial copper fungicide) commonly used to control *B. cinerea,* which can be higher than 500 mg L^−1^ [39]. However, the synergistic combinations obtained in this study could be an adequate alternative to conventional fungicides or be more efficient than another kind of CuNPs since a reduced copper amount (using NCuO) could inhibit the fungal growth of *B. cinerea*. In this sense, there are different parameters about nanoparticle characteristics that should be evaluated, such as size, copper oxidations, or the susceptibility of different strains of the fungus.

### 2.3. Synergistic Combined Treatments inducing Cellular Leakage on B. cinerea and F. oxysporum

The induction of cytoplasmic leakage of *B. cinerea* and *F. oxysporum* exposed to synergistic combinations was evaluated on a 6 h assay (Figure 3), in order to find possible mechanisms associated with the antifungal effect. As observed in Figure 3a,c, the release of protein content in *B. cinerea* was notably higher in combined treatments of NCuO-TEB and NCuO-IPR compared to each compound individually. This effect was more pronounced for the combination NCuO-TEB, where the protein content increased linearly in a time-dependent manner, 8-fold higher compared with control after 6 h. Thus, the impact on membrane integrity could be the main antifungal mechanism involved in the synergistic impact of NCuO-TEB and NCuO-IPR combinations on *B. cinerea*. The release of nucleic acids followed the same tendency (Figure 3b); however, in the case of NCuO-IPR (Figure 3d), the effect caused by the combination NCuO-IPR was similar to the effect caused by each agent individually, suggesting that the synergistic mechanism could be related to an impact at the membrane level instead of the nucleic level.

The same effect on protein content was observed for *F. oxysporum* (Figure 3e). However, the difference between the effect caused by the NCuO-TEB combination and the control treatment was less than the effect observed on *B. cinerea*. Contrastingly, the effect was more notable on the release of nucleic acids, particularly after 2 h (Figure 3f).

The synergistic effect is expected when two antifungal agents target different parts of the cell, while the additive effect occurs when the two agents target the same part [40]. Tebuconazole is known to regulate the permeability of fungal cell membranes via inhibition on ergosterol synthesis, which is a lipid constituent of microorganisms and fungi [41]. On the other hand, copper has been associated to the forming of reactive oxygen species (ROS) and hydroxyl radicals, interfering in the catalysis of Fenton and provoking changes on the activator protein 1 (AP-1) which regulates gene expression [42]. This could be an explanation of the observed synergistic effect on the membrane damage and the consequent leakage of the protein content. Other antifungal mechanisms not evaluated in this study could be involved on the synergistic effect on *B. cinerea* and *F. oxysporum*, considering the different resistance these two fungal genres can have to fungicides due to morphological differences [43].

Further chemical and biological investigation are needed to accurately clarify the action mechanisms involved between NCuO and fungicides, to inhibit fungal growth.

## 3. Materials and Methods

### 3.1. Chemicals and Microbial Strains

Analytical standards (>97% purity) of Iprodione (IPR), Tebuconazole (TEB) and Pyrimethanil (PYR) were purchased from Sigma-Aldrich (St. Louis, MO, USA). The copper oxide nanoparticles (NCuO) were purchased from Sigma-Aldrich (St. Louis, MO, USA, 99.99% trace metal basis, <50 nm particle size). NCuO were characterized by X-ray Diffraction (XRD, shown in Appendix A). As shown, peaks associated with copper oxide were observed at 2θ = 32.9°, 35.9°, 39.1°, 49.1°, 58.6°, 61.9°, 66.5°, 68.4°, 72.8°, and 75.5°, indexed as (110), (−111), (111), (−202), (202), (113), (311), (221), (311), and (222), corresponding to crystallographic planes of CuO powder matched to reference of the Joint Committee of Powder Diffraction Standard (JCPDS file no. 01-080-0076). The phytopathogenic fungi *Botrytis cinerea* (CCCT21.01) and *Fusarium oxysporum* (CCCT 21.02) were provided by the Chilean Culture Collection of Universidad de La Frontera (CCCT-UFRO) and cultured on Potato Dextrose Agar (PDA) for 7 days at 25 °C in darkness and 70% humidity. The assays were also performed using potato dextrose broth (PDB). Both culture media were purchased from Merck (Darmstadt, Germany).

### 3.2. Antifungal Activity

#### 3.2.1. Fractional Inhibitory Concentration

The fractional inhibitory concentration (FIC) was determined to find the type of interaction between NCuO and three fungicides: IPR, TEB and PYR, against *Botrytis cinerea* and *Fusarium oxysporum*. A checkerboard dilution assay was performed on a 48-well plate according to the test proposed by Hsieh et al. [44]. First, spore suspensions of *B. cinerea* and *F. oxysporum* were collected from the surface of each fungus colony grown on a PDA plate for 7 days, by scraping off with a sterile slide and 20 mL of a solution containing Tween 80 (0.05% *w*/*v*). These suspensions were filtered through four layers of sterile gauze and the spores were counted on a hemacytometer. Thus, aliquots of the spore suspensions in PDB were transferred to a 48-well plate to obtain a final concentration of 10^5^ spores mL^−1^ in presence of the treatments: NCuO (10–160 mg L^−1^ for both fungi), TEB (0.0375–2.4 mg L^−1^ for *B. cinerea*; 0.39–25 mg L^−1^ for *F. oxysporum*), PYR (10–640 mg L^−1^ for both fungi), and IPR (3–192 µg L^1^ for *B. cinerea*; 12–768 µg L^1^ for *F. oxysporum*). These treatments were evaluated individually and in combinations, and the concentrations were selected based on preliminary experiments. The plates were incubated for 3 days at 25 °C and 80 rpm to avoid nanoparticle aggregation. Triplicates of the plates were performed for each NCuO–fungicide interaction. The lowest NCuO and fungicide concentration inhibiting the mycelial growth was considered as the MIC. Then, the FIC was calculated according to Equation (1) proposed by Hsieh et al. [44]:(1)$$\frac{A}{M{IC}_{A}}+\frac{B}{M{IC}_{B}}=F{IC}_{A}+F{IC}_{B}=FIC$$
where A and B correspond to the MIC of NCuO and the fungicide when used in combination, whilst MIC_A_ and MIC_B_ correspond to their MIC individually. The interaction type was categorized according to FIC as synergistic (≤0.5), additive (0.5–1), or indifferent (1–4).

#### 3.2.2. Inhibition of Mycelial Biomass

Once the concentrations that provoked synergism were found (FIC ≤ 0.5) from the Section 3.2.1, it was necessary to corroborate the extent to which these treatments can inhibit the mycelial biomass growth of *B. cinerea* or *F. oxysporum* in a longer-term experiment. The method of Hou et al. was followed [45]. Briefly, mycelial disks of 5 mm (3 disks per flask) were obtained from the edge of the plate (grown for 7 days) and inoculated in Erlenmeyer flasks containing the synergistic combinations dissolved in PDB. Thus, two synergistic combinations were the treatments evaluated on *B. cinerea*: 40 mg L^−1^ NCuO + 0.3 mg L^−1^ TEB, and 20 mg L^−1^ NCuO + 12 µg L^−1^ IPR. One treatment was evaluated on *F. oxysporum*: 20 mg L^−1^ NCuO + 1.56 mg L^−1^ TEB. The corresponding MIC of each agent individually was also considered for comparison. After 10 days of culture at 25 °C and 120 rpm, the mycelial biomass was washed three times with sterilized water, then filtered to remove the liquid using a vacuum pump, and weighed after 24 h of drying at 40 °C. All the treatments were performed in triplicates. One-way analysis of variance (ANOVA) was determined to evaluate differences between the treatments. A Tukey’s HSD test was performed when significant differences were obtained (*p* ≤ 0.05) using SPSS 17 software (SPSS, Inc., Chicago, IL, USA) (Trial version). The synergism was corroborated according to the Abbott method using Equation (2), where the %CI_exp_ corresponds to the expected combined inhibition, and I_a_ and I_b_ represent the inhibition percent of NCuO and the fungicide, respectively. Secondly, the synergy factor (SF) was calculated according to Equation (3), where I_ab_ represents the obtained inhibition percentage of NCuO combined with the fungicide [46,47]. Herein, SF values ≥1 were considered synergistic, additive for SF values between 0.75 and 1, and antagonistic for SF values under 0.75. While the Hsieh equation in Section 3.2.1 was used to confirm whether synergism exists, the Abbot equation allows us to measure the degree of synergism based on the expected and the observed antifungal effect.
%CI_exp_ = I_a_ + I_b_ − (I_a_ × I_b_/100)(2)
SF = I_ab_/(%CI_exp_)(3)

#### 3.2.3. Cytoplasmic and Nucleic Acid Leakage

In order to elucidate the antifungal mechanisms involved in the synergistic interaction between NCuO and the fungicides, the cellular leakage from *B. cinerea* and *F. oxysporum* was evaluated by determination of soluble proteins and nucleic acid contents. The method described by Fernandez-San Millan et al. [48] was used with some modifications. In detail, spore solutions of both fungi containing 10^5^ cells/mL were cultured in PDB for 72 h. Then, the obtained mycelium was exposed to the treatments corresponding to the synergistic combinations of NCuO and fungicide selected from the Section 3.2.1. These treatments were incubated for 4 h and then, the pellet was centrifuged at 4000× *g* for 20 min. The supernatant was used to measure the leakage of nucleic acids by the optical density at 260 nm (OD_260_), according to Feng et al. [49]. Also, the intracellular soluble protein content was determined through the Bradford assay [50].

## 4. Conclusions

Copper oxide nanoparticles (NCuO) have been effective as antimicrobials against phytopathogens; however, high concentrations are required to inhibit fungal growth effectively. From the novelty perspective, we demonstrated that combining NCuO with fungicides can be a more efficient strategy than using each agent alone: the necessary amount of NCuO to inhibit the mycelial biomass of *Botrytis cinerea* and *Fusarium oxysporum* could be reduced 4- and 8-fold, which occurred by a potent synergistic interaction between NCuO combined with iprodione and tebuconazole (NCuO-IPR and NCuO-TEB).

Further studies should be addressed to evaluate how these combinations could be applied in practice or in field conditions. The development of stable nanoformulations based on NCuO-IPR and NCuO-TEB combinations can be a feasible strategy for the sustained release of both agents, without the necessity of repeated applications of pesticides, as is commonly performed by farmers. The application strategy will depend on the type of phytopathogen. For instance, *F. oxysporum*, as a soilborne fungus, could be controlled in soil by a nanoformulation capable of releasing NCuO and fungicides under specific conditions for fungal development; in the case of *B. cinerea,* a nanoproduct applied directly in the plant could prevent the gray mold in crops postharvest. Similarly, more fungal strains should be evaluated to enable a wider use of these combinations in agricultural applications.

Along with its effectiveness as an antimicrobial, the combined strategy of NCuO and fungicides could also help to alleviate the non-target toxicity produced by metals and pesticides by reducing the presence of their residues in the environment. However, this needs to be confirmed. As a risk assessment, further studies should evaluate the potential adverse effects of NCuO combined with fungicides on beneficial microbial cells in soil and their fate in the environment. It is important to mention that one of the main limitations of applying NCuO in field conditions is that regulations on using nanomaterials in agriculture are still under evaluation.

## Figures and Tables

**Figure 1 antibiotics-13-00215-f001:**
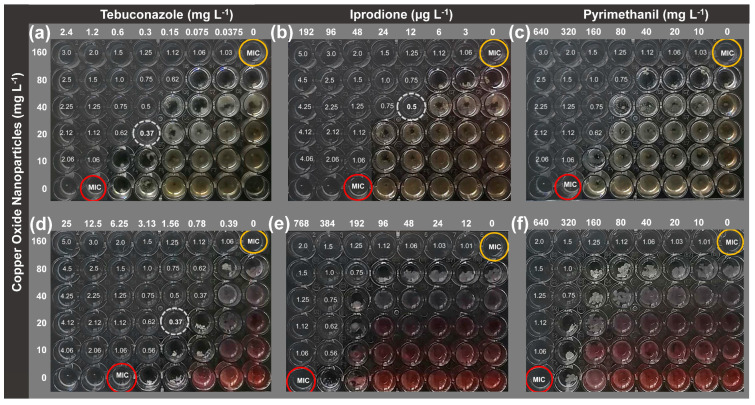
Checkerboard assay performed with NCuO combined with three fungicides: Tebuconazole, Iprodione, and Pyrimethanil against *Botrytis cinerea* (**a**–**c**) and *Fusarium oxysporum* (**d**–**f**). The values in the wells represent the calculated FIC in the combinations where fungal growth was absent. The yellow and red circles highlight the NCuO and fungicide MIC, respectively. Dashed circles highlight the FIC of synergistic interactions (i.e., <0.5).

**Figure 2 antibiotics-13-00215-f002:**
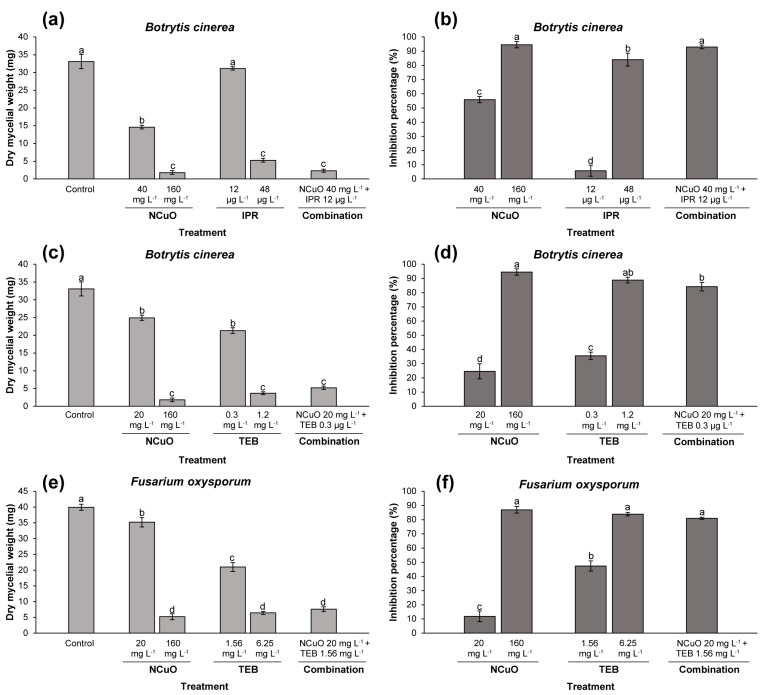
Effect of the NCuO combined with fungicides (synergistic concentrations) on *B. cinerea* (**a**–**d**) and *F. oxysporum* (**e**,**f**) mycelial biomass. The effect of NCuO and fungicide used alone are also included with their respective MIC. The inhibition was calculated as dry mycelial weight and as inhibition percentage. The results are presented as the mean ± the standard deviation of the triplicates. Different letters above the bars represent different statistical groups by Tukey-HSD comparisons at *p* ≤ 0.05.

**Figure 3 antibiotics-13-00215-f003:**
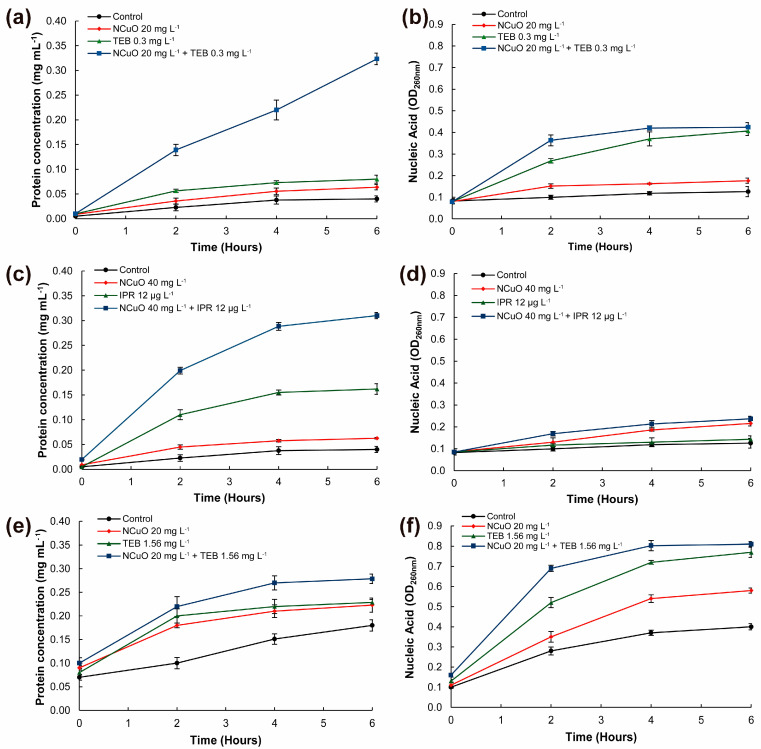
Synergistic concentrations of NCuO combined with Tebuconazole and Iprodione inducing cellular leakage on *Botrytis cinerea* (**a**–**d**) and *Fusarium oxysporum* (**e**,**f**) mycelia. Soluble protein (**a**,**c**,**e**) and nucleic acids (**b**,**d**,**f**) were determined from filtered solutions.

**Table 1 antibiotics-13-00215-t001:** Obtained inhibition (%) of NCuO (I_a_), fungicides (I_a_), their synergistic combinations (I_ab_), and the expected combined inhibition (Ci_exp_) against *B. cinerea* and *F. oxysporum.* These values were used to determine the synergy factor according to the Abbot method.

Synergistic Combinations	Inhibition (%)				Synergy Factor
	Alone Compounds		Combined Compounds		
	NCuO (I_a_)	Fungicide (I_b_)	Obtained (I_ab_)	Expected (Ci_exp_)	
Against *Botrytis cinerea*					
NCuO 20 mg L^−1^ + 0.3 mg L^−1^ TEB	24.6 ± 5.2	35.5 ± 2.6	84.3 ± 3.1	51.4	1.63
NCuO 40 mg L^−1^ + 12 µg L^−1^ IPR	55.8 ± 2.1	5.8 ± 3.8	93 ± 1.3	58.4	1.59
Against *Fusarium oxysporum*					
NCuO 20 mg L^−1^ + 1.56 mg L^−1^ TEB	11.8 ± 3.7	47.41 ± 3.5	80.9 ± 0.7	53.6	1.5

## Data Availability

Data are contained within the article.

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
