# Peer review of "Potential Antifungal Effect of Copper Oxide Nanoparticles Combined with Fungicides against *Botrytis cinerea* and *Fusarium oxysporum"

_antibiotics, 2024, doi:10.3390/antibiotics13030215_

Round 1
Reviewer 1 Report
Comments and Suggestions for Authors The manuscript submitted for review describes the synergistic effect of copper oxide nanoparticles and two known fungicides. The results obtained are interesting, but I have a few questions regarding the methodology of the experiments: 1. The authors state that the number of conidia was calculated using a hemacytometer. I must point out that using this method it is impossible to distinguish living cells from dead cells. 2. What was the accuracy of biomass weighing? 3. This work lacks studies showing that the tested "biocide systems" and copper oxide nanoparticles are safe for other 'beneficial' microbial cells living in the soil. 4. Authors should demonstrate the true novelty of this work.
Reviewer 2 Report
Comments and Suggestions for Authors
In Article “Potential antifungal effect of copper oxide nanoparticles combined with fungicides against two phytopathogenic fungi.” reported the SAR study of copper oxide nanoparticles (NCuO) is as an alternative to pesticides due to their reported antifungal effect against a great variety of phytopathogens. The combined use of NCuO with fungicides represents a more advantageous strategy to reduce the necessary amount of both agents to inhibit the fungal growth, simultaneously reducing their environmental release. This study aimed to evaluate the antifungal activity of NCuO combined with three fungicide models separately: Iprodione (IPR), Tebuconazole (TEB), and Pyrimethanil (PYR) against two phytopathogenic fungi: Botrytis cinerea and Fusarium oxysporum. The use of these pesticides to save the environment from toxicity.
The following minor corrections are recommended.
The cited reference must be in the same format and styles, in reference number 19 and 21. The published articles year required in a bold format. Check all references and correct them.
Author Response
Response to Reviewer 2 Comments
Summary: Thank you very much for taking the time to review this manuscript. Please find the detailed responses below and the corresponding revisions/corrections highlighted/in track changes in the re-submitted files.
Point by point response
Point 1: The cited reference must be in the same format and styles, in reference number 19 and 21. The published articles year required in a bold format. Check all references and correct them.
Response 1: All the references were checked and re-ordered according to changes produced.
Reviewer 3 Report
Comments and Suggestions for Authors
Comments to Authors:
1. Line 23: provide unit to the value.
2. In introduction: It will be nice to add costing perspective of both antifungal agents and Copper oxide nanoparticles.
3. Line 220-224: The above text should be shifted to results and discussion section. “As shown, peaks associated to copper oxide were observed at 2θ= 32.9°, 35.9°, 39.1°, 49.1°, 58.6°, 61.9°, 66.5°, 68.4°, 72.8° and 75.5° indexed as (110), (-111), (111), (-202), (202), (113), (311), (221), (311) and (222), corresponding to crystallographic planes of CuO powder matched to reference of the Joint Committee of Powder Diffraction Standard (JCPDS file no. 01-080-0076).”
4. Line 287: specify xg instead of rpm.
5. Line 289-290: by using kit? If yes, then provide kit details.
6. It will be nice to add some results by Microscopy either SEM or TEM.
7. Provide a note on nanoparticles application strategy in the field.
8. Provide the limitations of present investigation.
Comments on the Quality of English LanguageNeed improvements
Reviewer 4 Report
Comments and Suggestions for Authors
Title: Potential antifungal effect of copper oxide nanoparticles combined with fungicides against two phytopathogenic fungi
This manuscript is well-written by the authors. I do believe that if they can improve the manuscripts following all comments. It might have a chance to publish in the journal.
Comments
1. Topic: Please rewrite. It is complicated. The authors should add the names of two phytopathogenic fungi in the title.
Abstract
2. Line 13: The reduction in the use of pesticides has been occurring in many years not last few years. Please modify.
3. The authors should reduce the sentences or words in the introduction in the abstract. There are 7 lines. Actually, 1-2 short sentences are enough.
4. Line 22-24: Please rewrite. It is complicated to the readers.
5. Line 25: Please delete the word “Thus”.
Introduction
6. I strongly suggest the authors to modify or re-write the introduction.
-The first paragraph: describe the importance and the problem of Botrytis cinerea and Fusarium oxysporum in the agriculture.
-The second paragraph: describe strategies to eradicate the phytopathogenic fungi. The author can describe the use of nanoparticles as an alternative strategy.
-The third paragraph: describe the synergistic effects of the nanoparticles containing the bio-active compounds plus some antifungal agents.
-The fourth paragraph: describe the objective of this study. Why the authors are interested in this study?
Results and discussion
7. Based on the synergistic effects, the values are calculated by the MIC. Please add the information on the MIC values of each compound against the pathogens. The authors did not describe about that.
8. Fig. 1 presents the synergistic interaction of the nanoparticles plus the fungicidal agents. By the way, the authors should present the results as a Table to report the interaction between the nanoparticles and each fungicidal agent. Although, the authors have presented some synergistic combinations in Table 1 but I suggest to add the other synergistic interaction such as additive, indifferent in the table.
9. Please add the information of the discussion. Try to compare the results (the author’s hypothesis) with other finding by other researchers.
Materials and methods
10. Line 226-227: Please replace the word by “cultured on Potato Dextrose Agar (PDA)”.
11. The authors performed FIC index to determine the synergistic effects of the compounds against the pathogens. The FIC index is calculated based on the MIC values. By the way, the authors did not describe the MIC’s protocol in the methods. Please add the information.
12. Do the authors perform statistical analysis? If yes, please add the information.
Conclusion
13. Please modify the conclusion. It should be summarized on the key finding.
References
14. The references of 2022,2021, and 2023 are suggested to be cited.
15. Please remove some old references.
Comments on the Quality of English Language
Please edit the English grammar.
Round 2
Reviewer 1 Report
Comments and Suggestions for Authors Thank you for introducing corrections to the manuscript.I suggest publishing the results in the form presented.
Congratulations.
Reviewer 3 Report
Comments and Suggestions for Authors
Please accept